# Tranquillizing Effect of *Passiflora incarnata* Extract: Outcome on Behavioral and Physiological Indicators in Weaning Pigs with Intact Tails

**DOI:** 10.3390/ani12020203

**Published:** 2022-01-15

**Authors:** Grazia Pastorelli, Valentina Serra, Lauretta Turin, Veronica Redaelli, Fabio Luzi, Sara Barbieri

**Affiliations:** 1Department of Veterinary Medicine, University of Milano, Via dell’Università 6, 26900 Lodi, Italy; grazia.pastorelli@unimi.it (G.P.); sara.barbieri@unimi.it (S.B.); 2Department of Biomedical, Surgical and Dental Sciences, University of Milano, Via della Commenda 10, 20100 Milano, Italy; veronica.redaelli@unimi.it (V.R.); fabio.luzi@unimi.it (F.L.)

**Keywords:** pig, intact tail, botanical, behavior, physiological indicator, reactivity, tail lesion

## Abstract

**Simple Summary:**

Post-weaning is the most critical phase in pig farming, characterized by efforts to ensure health, performance and welfare of animals. Despite that EU Directive 2008/120/EC prohibits the practice of tail docking, it is still commonly applied in intensive farming to avoid tail biting. From a nutritional perspective, the dietary supplementation with natural extracts with calming properties could represent a promising approach to overcome common production stressors, reducing abnormal behaviors such as tail biting. This study intended to determine the effects of the dietary inclusion of *Passiflora incarnata,* known for its tranquillizing activity, on skin lesions, thermal imaging, behavior, salivary cortisol and IgA levels on post-weaning piglets reared with intact tails. Growth performances were also monitored. No differences were recorded between diets regarding growth performance, whereas findings concerning aggressive and abnormal behaviors, such as tail and ear biting, and lower levels of cortisol confirmed the hypothesis of the calming effect of *P. incarnata* on post-weaning piglets.

**Abstract:**

Tail docking has been used in the pig industry to decrease the occurrence of tail biting behavior. This abnormal behavior has a multifactorial origin since it is a response to simultaneous environmental, nutritional and management changes. Given the calming properties of *Passiflora incarnata*, we hypothesized that dietary supplementation with the extract in weaned pigs could result in a modification of behavior and physiologic indicators linked to stress. Weaned piglets (*n* = 120, mean body weight 9.07 ± 2.30 kg) were randomly allocated to one of two dietary treatments: control diet (CON) and CON supplemented with 1 kg/t of *P. incarnata* (PAS). The trial was 28 days long. The presence of skin lesions was assessed at d-1, d-10, d-19, and d-28, and saliva samples were collected for IgA and cortisol determinations at the same sampling times. Results showed the PAS group was characterized by equal growth performance as the CON group, fewer ear lesions (*p* < 0.05), less aggressive behavior (*p* < 0.001), higher enrichment exploration (*p* < 0.001) and lower cortisol levels (*p* < 0.01). Time effect was observed for tail lesions (*p* < 0.001) and behavioral observations (*p* < 0.001). Additional research is required to determine the effect of *P. incarnata* extract using a larger number of animals and longer period of supplementation when risks associated with tail biting are uncontrolled.

## 1. Introduction

Tail docking has been used in the pig industry to lower the occurrence of tail biting behavior. Tail docking reduces the potentially dangerous associated effects (pain and infection), but it does not eliminate the inner or outer causes that trigger the motivation of tail biting [1]. In addition, tail docking is a painful procedure with short-term consequences, such as procedural pain [2], and long-term effects, which may include formation of traumatic neuromas [3]. Due to those welfare considerations, tail docking should be carefully assessed as a common husbandry practice. According to the Council Directive 2008/120/EC [4], tail docking should not be applied regularly in the European Union (EU), but it is allowed only if the chance of tail biting is assessed and the evidence of tail injuries persists after the application of improvement procedures.

To avoid tail biting as well as tail docking, and to help farmers comply with current legislation—maintaining both welfare and productivity—in 2016 the European Commission enacted a recommendation concerning the general requirements to avert tail biting and minimize routine tail docking [5]. This recommendation asserts that the EU Member States have to make sure that farmers perform tail biting risk evaluations and that, according to the outcome, proper modifications in housing and management conditions are considered. The Commission also called for Members States practical plans to eliminate routine tail docking, including the gradual introduction of pigs with intact tails in farms applying a number of practical prevention measures.

Tail biting has a multifactorial origin since it results from the simultaneous presence of multiple causes. Multiple causes have been linked to increased risk of tail biting, encompassing overcrowding, power environment, bad air quality, mediocre health status [6] and poor-quality diets. Pigs in contemporary farming systems are subjected to many stressful conditions, which cause a negative impact on health and welfare [7,8]. Stress to weaning pigs is mainly caused by the combination of separate litters, transportation to different areas, housing settings, nutrition and major changes in diet. Such external factors have been implicated in tail biting as well as internal factors and personality traits, such as fearfulness and anxiety [9].

From a nutritional point of view, a diet spiked with tryptophan resulted in decreased activity and aggressiveness and increased lying behavior in grower gilts [10]. Contrariwise, supplementation with ractopamine, which augments activity in pigs, is correlated to enhanced oral–nasal behavior and aggressiveness [11]. Few studies demonstrated welfare benefits, most likely occurring from diminished negative behaviors, due to supplementation with magnesium in the pig diet, which appears to have calming effects [12,13,14].

Another dietary approach could be the integration with natural extracts that have different biological properties. In this paper, *Passiflora incarnata* L. was examined, as its calming effect is known in humans. This plant, classified in the *Passifloriaceae* family and habitually considered as passion fruit, is spread in tropical regions of the world, and its extract is commonly utilized to treat anxiety, erethism, neuralgia and insomnia in Europe and South America [15]. Its recognized calming effects have also been proven in experiments on rats and mice [16,17]. Lately, it was demonstrated that it could be utilized as a promising source of antioxidant and anti-inflammatory compounds [15]. The major phyto-constituents of *P. incarnata* are flavonoids, which are concentrated in leaves and include apigenin, luteolin, quercetin, kaempferol, C-glycosyl flavonoids, vitexin, isovitexin, orientin, isorientin, schaftoside, isoschaftoside and swertisin [18].

There are limited studies on the use of *P. incarnata* in animals, and in particular in pigs. Its supplementation in slaughter pigs [14] and growing pigs [19] did not exert any detrimental effect in terms of carcass and meat quality parameters, also reducing anxiety and irritability in pigs during transport. To the best of our knowledge, its use in post-weaning piglets has only been tested in a previous study with the same level of inclusion [20], showing no adverse effects. This additive complies with the Register of Feed Additives (Annex 1 Released date 6 December 2021).

Considering the correlation between physical activity and establishment of abnormal behaviors [21], and given the therapeutic properties of *Passiflora*, we hypothesized that dietary supplementation with this natural extract in weaned pigs could result in a modification of behavioral mechanisms unevolved in the onset of abnormal behaviors, such as tail biting and in improvement of physiologic indicators linked to stress.

## 2. Materials and Methods

### 2.1. Animals and Treatments

The study was performed on a commercial pig farm located in northern Italy. A total of 120 Goland hybrid piglets (40 days old) with intact tails, 60 castrated males and 60 females, with a mean body weight of 9.07 kg ± 2.30, were selected from 11 litters of contemporary sows. Piglets were individually ear-tagged and divided into two experimental dietary groups (5 pens per diet, 12 piglets per pen), balanced for sex and body weight. Each group was assigned to a pen (4.13 × 1.45 m) with partially slatted floor and equipped with a self-feeder and two nipple drinkers to allow ad libitum access to feed and water during the 28-day experimental period. Rooms had a forced-air ventilation system set at 60% relative humidity and 24 °C temperature. Metal chains with soft wooden bars hanging from the walls were routinely supplied in each pen as enrichment material.

Animals were assigned to two dietary treatments: a control diet (CON) and diet supplemented with 1 kg/t of *Passiflora incarnata* (PAS) titrated 3.5% total flavonoids calculated as vitexin (Table 1). The supplement included a water-soluble extract of *P. incarnata* L. (*Passifloraceae* spp.). Flowering aerial parts were prepared on an industrial scale by a standardized procedure that involved ultrasonic extraction with 70% ethyl alcohol and 30% water followed by spray-drying with maltodextrins and silicon dioxide as excipients. As reported by the certificate of analysis supplied by the retailer (Farmacia Bonfanti, Salsomaggiore Terme, PR, Italy), quantitative analysis of the phenolic compounds was carried out by HPLC-UV-DAD (high-performance liquid chromatography, UV diode array detector). Analytical methods complied with current edition of the European Pharmacopoeia.

The diets were formulated to meet the nutritional needs of a weaned piglet [22]. Individual piglet weights (at day 0 and day 28) and pen feed consumption were recorded, and the average daily gain (ADG) and feed conversion ratio (FCR) were calculated. Feed conversion ratio (FCR) was calculated by dividing the amount of feed consumed during the experimental period by the growth of the animals during that same time. Average daily gain was calculated from measurements of weight and number of experimental days. Feed intake of the pen was calculated by the difference between offered feed and leftovers. The leftovers were weighed daily, if any, and not considered for final calculation of feed consumed. Mortality was recorded daily throughout the trial.

The protocol was approved by the Animal Welfare Committee of the University of Milan (OPBA_157_2019), according to the Directive 2010/63/EU. The experiment was run under adequate housing and management conditions. In case of an outbreak of tail biting, routine management on the farm established that pigs with severe wounds would receive appropriate veterinary treatment and, according to veterinarian advice, would be removed from the pen. All pens with cases of tail biting would be provided with additional enrichment material to maintain tail biting within an acceptable level.

### 2.2. Health Physiological Indicators

#### 2.2.1. Thermal Imaging

The skin temperatures were recorded on the dorsal, ocular, front ear and back ear regions using an Avio thermoGear Nec G120EX microbolometer infrared camera (Nippon Avionics Co., Tokyo, Japan) (320 × 240 pixels). Data were acquired four days after the creation of groups (day 4), then at day 11, day 17 and day 24 by an operator standing at the entryway of each pen at about 2 m from the animals and at a height of 1.60 m. Thermal images were collected from six randomly selected piglets per pen. The camera was automatically calibrated prior to any image, and emissivity was set at 0.96. Thermal images were downloaded and analyzed using NEC InfRec Analyzer and Grayess IRT Analyzer software (Nippon Avionics Co., Ltd., Tokyo, Japan). The skin temperatures in each area were extracted as shown in Figure 1.

#### 2.2.2. Collection of Saliva Samples

The day after the creation of groups (day 1) and every nine days (day 10, day 19 and day 28), saliva samples were collected from five randomly picked piglets per pen, for a total of 200 samples. Saliva samples were collected using Salivette^®^ tubes (Sarstedt, Aktiengesellschaft & Co., Nümbrecht, Germany) containing a cotton sponge. The piglets were allowed to chew on the sponge, which was clipped to a flexible thin metal rod, for at least 30 s until thoroughly moist. Tubes were maintained on ice until arrival at the laboratory and then centrifuged at 3500× *g* at 4 °C for 10 min to obtain saliva. Samples were stored at −20 °C until the day of analyses.

#### 2.2.3. Salivary Analyses: IgA and Cortisol Evaluation

In this study, salivary IgA and cortisol were quantitated by the following assays. IgA concentrations were determined using a commercial enzyme-linked immunosorbent assay (ELISA; Pig IgA ELISA kit, Bethyl Laboratories, Inc., Montgomery, TX, USA) based on a sandwich binding, in which IgA molecules present in saliva samples are captured by anti-pig IgA antibodies that have been pre-adsorbed on the surface of microtiter wells. The colorimetric reaction, catalyzed by streptavidin-conjugated horseradish peroxidase, produced a yellow product, which was proportional to the amount of IgA analyte present in the sample. Saliva samples were analyzed in duplicate at a dilution of 1:500. Range of detection for this ELISA was 1.37–1000 ng/mL. Cortisol levels were measured by competitive ELISA (Salivary Cortisol Assay Kit, SLV-2930, DRG Instruments, GmbH, Marburg, Germany) in the same saliva samples. The microtiter wells were coated with a monoclonal (mouse) antibody directed towards an antigenic site on the cortisol molecule. Endogenous cortisol present in saliva samples competed with a cortisol-horseradish peroxidase conjugate for binding to the coated antibody. Since the amount of bound peroxidase conjugate was inversely proportional to the concentration of cortisol in the sample, the intensity of color was inversely proportional to the concentration of cortisol in the sample. For this assay, saliva samples were analyzed undiluted in duplicate. Range of detection for this ELISA was 0.1–30 ng/mL. The absorbance values of both assays were measured with the spectrophotometer SpectraMAX 340PC (Molecular Devices Corporation, San Jose, CA, USA) at a wavelength of 450 nm.

#### 2.2.4. Skin and Tail Lesions

The appearance of skin and tail lesions was evaluated on each animal by one trained observer using a validated scoring system defined in the Welfare Quality^®^ [23]. Data were collected on the day after the creation of groups (day 1) and every nine days (day 10, day 19 and day 28). Skin lesions were scored as follows: “score 0” up to 4 lesions; “score 1”, from 5 to 10 lesions; “score 2” more than 10 lesions. The lesions were assessed in 4 body regions—ear, front, middle and hind-quarters—by visually assessing one side of the piglet’s body from inside the pen. Tail lesions were scored using a 0 to 2 scale as follows: “score 0” animals without lesions; “score 1” animals with superficial lesions along the length of the tail, without evidence of fresh blood and swelling; “score 2” animals with lesions characterized by fresh blood, swelling and infection, or lack of part of the tail with or without the presence of coagulated blood.

### 2.3. Behavioral Indicators

#### 2.3.1. Behavioral Observations

The behavior of pigs was recorded from four days after the beginning of the experimental feeding period up to the end, using a digital video recorder (Panasonic, WJ-HL208, Panasonic, Tokyo, Japan) and high-definition day/night cameras (Panasonic, WV-CP500, Panasonic, Tokyo, Japan). Behavioral observations were conducted by a single observer on four periods of recorded footage. Each period consisted of three continuous days (from 9.00 a.m. to 12.00 a.m.) selected at regular intervals and not including the days of other tests or farm interventions: Period 1 (day 7–9), Period 2 (day 13–15), Period 3 (day 20–22) and Period 4 (day 25–27). 

Behaviors were assessed using all occurrence sampling, to continuously record the selected behaviors in the group [24]. Behavioral categories were defined as follows: aggressive interaction (fight or displacement with or without physical contact; attitudes of threat and submission were not considered); explorative interaction (all specific behavioral elements such as rooting, sniffing, biting and chewing) towards (i) pen components, (ii) pen mates (all parts of the body of the other pigs, with the exception of tails and ears); (iii) tail; (iv) ear.

#### 2.3.2. Novel Object Test

Both groups of piglets were subjected to the Novel Object Test (NOT) 3 times at 9-day intervals (day 10, day 19 and day 28) in the home environment. The order of testing was balanced for pen position in the room and treatment, and no adjacent pen was tested one below the other.

A novel object was presented to the pigs in the form of three different plastic objects at different time points: day 10—black plastic bucket; day 19—blue plastic drawer; day 28—orange traffic cone. The objects were lowered inside the pen from outside without disturbing or isolating the animals. The observer was placed outside the pen to have a full view of the novel object and animals without having to enter it. For the NOT, the focus of interest was on the degree of intensity of interaction with the novel object: standing (standing with head upward and ears pricked), weak interacting (exploring the novel object by gently nosing or sniffing), and strong interacting (exploring the novel object by rooting, chewing, pinching or pushing). The objects were left in the pen for the total time span of the behavioral test (33 min), and the total number of interactions was recorded at 30 s and 1, 2, 3, 31, 32 and 33 min from the time the object was introduced.

### 2.4. Statistical Analyses

Statistical analyses were performed with SPSS software (SPSS/PC Statistics 26.0, SPSS Inc., Chicago, IL, USA). Prior to hypothesis testing, all data were analyzed for normality and transformed where appropriate. Performance data were analyzed using the ANOVA procedure with dietary treatment and sex as main effects, and the pen was considered as the experimental unit. Cortisol changes over time were assessed by the Friedman test, while the Mann–Whitney U test was utilized to assess dietary treatment effect. Salivary IgA and behavioral data were logarithmically transformed (log_10_) to meet the normality assumption; then they were analyzed by analysis of repeated measures and General Linear Model, respectively. For both salivary data analyses, the animal was considered as the experimental unit. Behavioral data were analyzed using the ANOVA procedure with dietary treatment and period as main effects, and the random effect of pen. NOT data were analyzed by two-way ANOVA, considering dietary treatment and time of recording as fixed effects. Skin lesions were analyzed with logistic regression; since a score of 2 for skin lesion was found just in one subject of the control group, in the ear region, the variable was dichotomized (0: no lesion; 1: presence of lesions) and bivariate probit analysis (PROC GLM) was applied. Data of tail lesions were analyzed using ordinal logistic analysis (PROC GLM). Models included dietary treatment, time of recording and their interaction as predictor variables. The distribution into the three severity classes was reported as relative frequency. Results are presented as odds ratios (OR) with the associated 95% confidence interval. Alpha level for determination of significance was 0.05 and from 0.05 to 0.10 for trends. Thermal imaging was analyzed by two-way ANOVA, considering dietary treatment and time as fixed factors. Differences between periods were assessed by the Tukey post-hoc test for multiple comparisons. For the thermal imaging and behavioral data, the pen was considered as the experimental unit.

## 3. Results

### 3.1. Growth Performance

Pigs (*n* = 2 in PAS group and *n* = 1 in CON group) with severe wounds were removed from the pen, and one pig died in the CON group at d-19. Table 2 summarizes the results of growth performance. There was no effect of dietary treatment on pig body weight, feed intake or average daily gain. Male pigs at the end of the trial weighted slightly more than females (22.73 vs. 22.26), with no significant difference.

### 3.2. Health and Physiological Indicators

#### 3.2.1. Thermal Imaging

Table 3 reports the descriptive statistics for surface temperature measured for dorsal, ocular, front ear and back ear regions.

#### 3.2.2. Salivary IgA

IgA data were acquired from five randomly selected piglets per pen, for a total of 200 samples. IgA concentrations did not show any significant difference both considering treatment (*p* = 0.27) and time (*p* = 0.23) effects, while a significant diet x time interaction was found (*p* < 0.001) (Table 4). 

#### 3.2.3. Salivary Cortisol

Concentrations of salivary cortisol at different times of recording are shown in Figure 2. Salivary cortisol values in CON group ranged between 2.46 ± 0.65 ng/mL and 5.14 ± 0.65 ng/mL, while in the PAS group the values ranged between 2.38 ± 0.67 and 4.35 ± 0.68 ng/mL. A significant difference (*p* = 0.008) between the two experimental groups (5.14 ± 0.65 ng/mL CON vs. 2.38 ± 0.67 ng/mL PAS) was observed at d-28, and a trend (*p* = 0.055) was reported for time.

#### 3.2.4. Skin and Tail Lesions

No lesions were detected in front and hind quarter regions. Considering the middle region, only three pigs per dietary treatment showed lesions with score 1, and no effect of diet (*p* = 0.99) or time of recording (*p* = 0.74) were detected. Piglets fed the PAS diet (OR = 0.18; 95% CI = 0.05–0.65) were less likely to have ear lesions compared to CON piglets (*p* < 0.05) (Figure 3). No effect of time of recording was detected (Figure 4).

All pens in the CON group and three pens in the PAS group reported at least one pig assessed as score 2 for tail biting during the experimental period. Those pens were provided with additional enrichment material. Tail lesions were not affected by dietary treatment (*p* = 0.973), and a time effect was detected (*p* < 0.001) (Figure 4).

### 3.3. Behavioral Indicators

#### 3.3.1. Behavioral Observation

The results of behavioral observations are reported in Table 5. Considering aggressive behavior, significant effects of dietary treatment and time (*p* < 0.001) and diet x time interaction (*p* < 0.05) were found. Considering exploratory behavior, no statistical significance was found for exploration of pen mates (diet *p* = 0.334; time = 0.093; diet x time interaction *p* = 0.136), while significant effects of time (*p* < 0.05), dietary treatment and diet x time interaction (*p* < 0.001) were found for manipulating the enrichment. Exploration of tails was significantly less observed in PAS (*p* < 0.001) and at the different times of recording (*p* < 0.001), while no effect of diet x time interaction was observed (*p* = 0.151). The effects of dietary treatment (*p* < 0.001), time of recording and diet x time interaction (*p* < 0.05) were found for exploration of the ear.

#### 3.3.2. Novel Object Test

The intensity of interaction with the novel object did not show any statistical difference between dietary treatments at different time points. Weak interaction was rarely observed in both groups; standing and strong interactions were more frequent 3 min after the object was introduced than 30 min after, as expected.

## 4. Discussion

A multidisciplinary perspective to the evaluation of welfare was applied in this study, which focused on health as well as behavioral and physiological indicators. In accordance with the above-mentioned EU recommendations regarding the need for reducing tail-docking, this research intended to examine the efficacy of *Passiflora incarnata,* known for its sedative and anti-anxiety properties, on the welfare of post-weaning piglets with intact tails. Published data indicate that the passionflower itself, along with its preparations, help to lower stress and may therefore be considered in the treatment of anxiety and restlessness [25].

### 4.1. Growth Performance

The results of growth performance agree with those reported in a previous study performed on post-weaning piglets whose diet was supplemented with the same natural extract at the same dosage [20]. Growth performance found herein can be evaluated as positive, considering that feed was not medicated and the mortality rate measured was very low. It should also be taken into account that the consumption of feed, which was found to be almost identical in the two experimental groups, highlights that the natural extract does not confer unpleasant odors or flavors capable of modifying the intake.

To the best of our awareness, there are few studies focused on the efficacy of *P. incarnata* on growth performance, especially in pigs. In the study conducted by Casal-Plana et al. [26], pigs supplemented with herbal derivatives of *Valeriana officinalis* and *P. incarnata* showed a significant rise in body weight in comparison to the control group at 22 and 24 weeks of age. The authors attributed the result to the tranquillizing effect of the herbal derivative, which by reducing aggressions, deviates energy from negative social behavior to growth [26]. Similar to our results, in the study conducted by Perondi et al. [27] the content of Passion fruit seed meal (*Passiflora edulis*) up to 16% did not impair the performance in growing and finishing pigs, since there was no impact on daily feed consumption, average daily gain and feed conversion.

### 4.2. Health and Physiological Indicators 

The interest for the evaluation of non-invasive health and physiological indicators and biomarkers to determine animal welfare is growing increasingly within the scientific community [28].

Thermography represents an appealing method, as it is a distant, noncontact and noninvasive procedure for collection of individual animal data [29]. In pigs, the ear skin temperature is a good measure of thermoregulatory responses [30], and thermography applied at the eye level represents a valuable method to estimate the core temperature of pigs under farm conditions [31]. In the present study, homogeneity of recorded temperatures seems in line with other results obtained [32] since no chronic stressors or morbidity was registered in the animals. The thermal results, considering the different locations with highest correlation with core temperature (such as ear, eye and dorsal region), may change with age, and this could explain the effect of time of recording in the present study.

Among these biomarkers, immunoglobulin A (IgA) is at the cutting edge because of its pivotal function in mucosal immunity and its relationship with physical health, stress and general psychological well-being.

The IgA amounts in normal healthy pigs are approximately 10–70 µg/mL [33]; the IgA concentrations found in this experiment showed values included in this range. IgA concentrations can be impacted by both physical and psychosocial stressors [28]. IgA has been recognized as a stress biomarker in rats [34], dogs [35] and humans [36]. In pigs, an augment in saliva IgA has been observed after immobilization [33,37], in endotoxemia [38] and after isolation [39]. In the case of isolation, IgA varied more than other markers such as cortisol and testosterone [39]. The lack of an increase in IgA in this case can be considered a positive result, underlying the absence of induced stress. Moreover, different response mechanisms are awaited when using IgA as a welfare marker. Chronic stressors may induce immune suppression and lower IgA, while acute sickness could be linked to augmented IgA in the contest of an immune response to cope with underlying pathologies. In the present paper, results seem to indicate no presence of chronic stressors from animals. 

Cortisol is a stress hormone synthesized by the adrenal glands and is often used as a biomarker to determine chronic stress in several animal species, including pigs [40]. Cortisol can be quantified primarily in plasma or serum, but other biological matrices such as saliva, urine and feces represent good alternatives [41,42,43]. Since blood sampling requires restraint of the pig during bleeding, and therefore causes stress, we chose to use saliva, which has the advantage of being a non-invasive sample. Moreover, salivary cortisol levels represent valuable indicators of circulating cortisol concentrations in pigs and reflect the biologically active, unbound cortisol in plasma [44]. 

In the present work, salivary cortisol concentrations measured in both groups were alike to those reported in several works on post-weaning piglets. For example, in the work conducted by Jarvis et al. [45], the cortisol values found in piglets weaned at various ages (12, 21 and 42 days) ranged from 1 to 5 ng/mL, showing higher cortisol values during the suckling phase, before the start of the weaning.

In a study conducted to investigate the effect of two environmental enrichment strategies (hanging objects and substrate) on post-weaning stress in piglets, salivary cortisol values ranged between 3.94 and 8.50 ng/mL [46]. The lowest cortisol values corresponded to the animals who had been provided with a wood bark substrate, suggesting a link between low cortisol concentration and decrease in stress due to the presence of environmental enrichments, which improved the piglets’ ability to adapt to the weaning stress in intensive farming.

In the present study a significant augment in cortisol level was found in the CON group at day 28. An augment in cortisol concentrations could also be associated to a lower on-farm animal welfare [47] and to other stress stimuli, such as transportation and temporary restraint, as observed in several studies conducted in pigs [48,49]. We speculated that the higher incidence of ear lesions could affect animal welfare resulting in a higher cortisol content in the CON group than the PAS group.Barnett et al. [50] observed that a lower aggressive behavior is linked to lower concentrations of cortisol in pigs, which could partially explain our findings. The higher cortisol levels observed at the end of the experimental period might be linked also to an underlying stressful situation that lasted in the CON group, despite the lower incidence of ear lesions, probably mitigated by farmer interventions.

Similarly, a reduction in salivary cortisol concentration has been observed in a group of undocked piglets after supplementation of the diet with a magnesium extract (known for its calming effect), compared to the control group (1.30 ± 0.09 ng/mL vs. 1.47 ng/mL) [51].

The correct timing for salivary sampling must be taken into consideration since baseline cortisol concentrations are lower in the evening. In this experiment, saliva samples have been collected at the same time (between 08:00 and 09:00), since basal levels of salivary cortisol go along with a circadian pattern of secretion. Since a peak generally appears in the morning, this is considered a good time to compare differences between experimental treatments.

Since the circadian rhythm of cortisol does not develop completely until 20 weeks of age [52], the determination of daily baseline cortisol concentration in young animals becomes unpredictable. In our study, a significant difference (*p* = 0.008) between the two experimental groups (5.14 ± 0.65 ng/mL CON vs. 2.38 ± 0.67 ng/mL PAS) was observed in correspondence of the last time point of sampling. Given the immaturity of the above-mentioned cortisol circadian rhythm, the lower concentrations of salivary cortisol detected in the last time point of sampling suggests that the dietary treatment may have exerted a role in reducing stress, since all the piglets enrolled in our study had an average age of 70 days at the end of the trial.

Risk factors for tail biting were checked, and enrichment material was provided during the study. Nevertheless, during the trial 21.86% of pigs reported tail lesions due to tail biting, of which 9.55% were severe lesions (score 2). A very low percentage of pigs was affected by lesions in the ear region (3.8%) and almost no lesions on the body (1.3%); no severe skin lesions were observed. No ear lesions could be referred to ear biting.

Tail biting is considered non-aggressive behavior [53], whereas body lesions are the result of agonistic behavior inside the group. Aggression in a stable group may occur because of competition for resources with limited access or other events, which disrupt the social stability. In the present study, feeding and nutritional requirements as well as space availability were guaranteed.

Dietary treatment reduced the probability to have lesions in the ears, resulting in lower percentage of piglets with ear lesions; the hypothesis of the calming effects of *Passiflora* extract may support the result obtained. Van Putten [54] argues that ears and tails are the easiest to chew, but ear chewing is more likely to provoke an attack by the recipient. In the present work, ear lesions occurred only in the CON group, likely due to the calming effect of *Passiflora* extract. A previous study [20] confirmed the calming effect of *Passiflora* extract in post-weaning pigs in which a lower percentage of scores occurred in the treated group.

The prevalence of tail lesions increased as the age of pigs progressed, in agreement with other studies [55]. Even if the time of recording stopped at 28 days, we agree with Calderón Díaz et al. [56] who speculated that as pigs get older, they are more capable of defending their ears from attention by others; therefore, biting pigs switch their attention towards the more easily accessible tail.

### 4.3. Behavioral Indicators

Aggression at mixing occurs to establish a dominance hierarchy, and post-weaning is one of the crucial phases related to animal welfare [57]; there are numerous attacks in the first weeks after weaning. Considering aggression, our results confirm data present in the literature [58], which report a higher number of attacks in the first two weeks. In both groups the same decreasing frequency over time was observed. Our results reported statistically significant differences in relation to treatment, as the aggressions were more frequent in the CON group compared to the PAS group.

These results concerning aggressive and abnormal behaviors, such as tail and ear biting, confirm the hypothesis of the calming effects of *P. incarnata* [20,26] on post-weaning piglets. The herbal extract may produce an effect on the activity level, reducing agonist interaction and exploration of tail and ear.

In consideration of the multifactorial nature of tail biting, providing good enrichment and observing how pigs interact with materials is crucial to fulfill their ethological needs, even if it may not be enough to prevent an outbreak. According to other authors, it is important to verify if pigs express and maintain exploration without losing interest in such enrichments over time [59]. For this reason, it is recommended to introduce enrichments or change them quite often during the most critical periods. Considering exploratory behavior towards enrichment, the number of interactions with the manipulable objects offered in the pen (hanged soft wood log and chain) was unexpectedly higher in the PAS group on almost all the observation days, even if for both groups the frequency decreased in the last two weeks. The marginal value of the provided materials [60] may have affected the level of exploration leading to diverging results from those connected to aggressive and abnormal behaviors.

Tail-biting outbreaks can appear quickly and are difficult to contain; therefore, the focus on minimizing risk factors should be sustained. It has been hypothesized that early detection of tail-biting behavior, before serious injuries appear, followed by management intervention could reduce the spread of this abnormal behavior. In the present study, exploratory behavior towards the tails was higher in the first two weeks than in the second half of the trial, while tail lesions were observed mainly in the following weeks as a probable result of the exploratory behavior. Our data may confirm that tail biting can be detected until lesions are present [61], demonstrating that manipulation of the tail may represent an early warning indictor for tail biting [62]. We should consider that, after the first lesions occur, the farmer removed pigs with severe wounds and treated tails with disinfectant solution that might have acted as repellent. These interventions may have altered the tail biting behavior and justify the decrease in such behavior.

The literature confirms that tail biters may show many other types of abnormal and harmful behaviors such as ear biting [63]. However, in the present study we did not observe signs of ear biting, but there were lesions due to aggressive behavior in the ear area. The farmer’s decision of treating tails with disinfectant solution may have led to the remission of tail biting, due to less exploratory behaviors towards the tail.

The outbreak of tail biting may be detected by changes in behaviors as well as by evaluating the response to reactivity tests.

Our results did not highlight any significant difference between the two experimental groups at different time points for the NOT. The decreased interest that has been observed in the present study 30 min after the introduction of the objects agrees with previous works.

The tendency to develop tail biting has been hypothesized to relate to pig behaviors in novelty tests [9]. Ursinus et al. [62] identified the pigs with the greatest interest in the new stimulus as the most biting areas; this fact could be interesting as this test could be used to identify “potentially biting” individuals. Our results do not show any difference in behavior connected to outbreaks of tail biting, and NOT did not represent a predictive measure for an early identification of tail biting.

## 5. Conclusions

Our research indicated that oral administration of *P. incarnata* extract results in unchanged growth performance and IgA level, underlying no presence of chronic stress or acute illnesses. A higher cortisol content was observed at the end of the trial in the control group, whereas a higher proportion of pigs affected by ear lesions was observed at the previous time point. This result might be linked to an underlying stressful situation that lasted during the second half of the experimental period, even showing differences in aggressive behaviors between groups. The general low presence of pigs with lesions, and thermal imaging results associated with adequate housing and management conditions, suggest a calming effect of herbal supplementation. The lowest level of exploration in PAS both for tails and ears suggests that the botanical compound may have an effect on the onset of abnormal behavior.

Additional studies are envisaged to determine the effect of *P. incarnata* extract using a larger number of animals and longer period of supplementation when risks associated with tail biting are uncontrolled.

## Figures and Tables

**Figure 1 animals-12-00203-f001:**
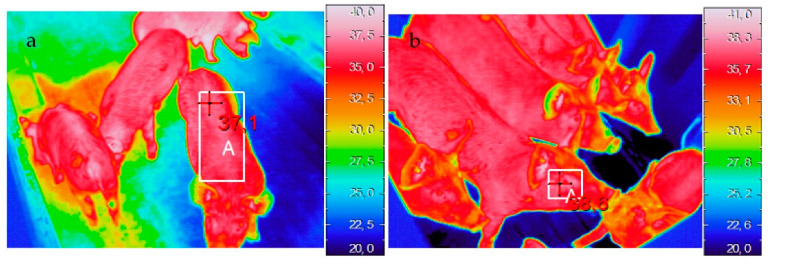
Examples of thermal images of the dorsal (**a**), ocular (**b**), front ear (**c**) and back ear (**d**) regions. The white rectangles indicate the areas in which the maximum temperatures were recorded.

**Figure 2 animals-12-00203-f002:**
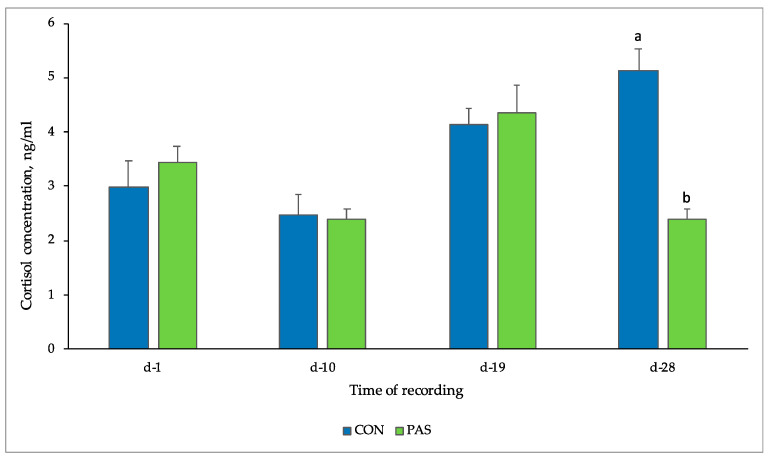
Cortisol concentrations (mean ± SE) found in saliva samples of two experimental groups. CON: control animals receiving only the basal diet without natural extract supplementation; PAS: animals receiving the control diet supplemented with *Passiflora incarnata* extract (1 kg/t); ^a,b^ Means within a row with different letters are significantly different at *p* < 0.05.

**Figure 3 animals-12-00203-f003:**
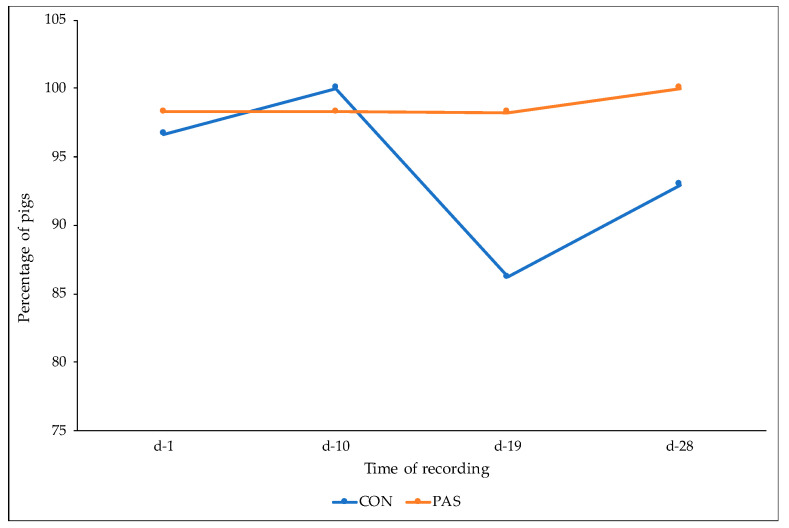
Proportion (%) of pigs from CON and PAS diets not affected by ear lesions (score 0) at different times of recording. CON: control animals receiving only the basal diet without natural extract supplementation; PAS: animals receiving the control diet supplemented with *Passiflora incarnata* extract (1 kg/t).

**Figure 4 animals-12-00203-f004:**
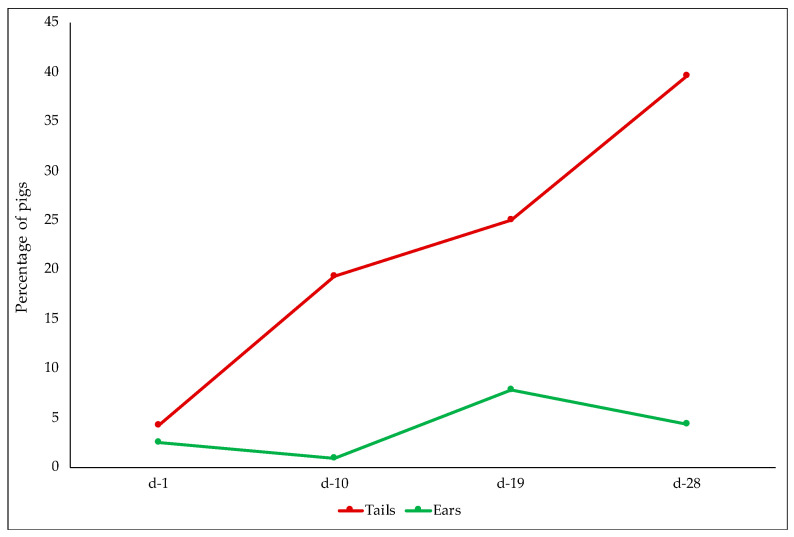
Percentage of pigs with tail and ear lesions at different times of recording.

**Table 1 animals-12-00203-t001:** Ingredients (%) and proximate chemical composition of the basal diet (as-fed basis).

Ingredients	%
Corn meal	29
Barley meal	25
Whey powder	12.5
Hulled barley	12.5
Soybean meal 48%	11.8
Wheat bran	6.0
Coconut oil	1.0
Dextrose monohydrate	1
Vitamin-Mineral premix ^1^	0.40
L-Lysine	0.50
Sodium chloride	0.20
DL-Methionine	0.18
L-Threonine	0.15
L-Tryptophan	0.07
**Chemical composition ^2^**	
Crude protein,	17.19
Ether extract	4.88
Crude fiber	3.16
Starch	40.90
Lactose	3.77
Lysine	1.24
Calcium	0.66
P dig	0.44
Net Energy (NE), kcal/kg	2462.44

^1^ Premix contained the following per kg of the diet: 15,000 IU vitamin A, 10 mg vitamin B1, 16 mg vitamin B2 (riboflavin), 2000 IU vitamin D (cholecalciferol), 250 mg vitamin E, 0.05 mg vitamin B12 (cobalamin), 2 mg vitamin K, 50 mg vitamin B5 (niacin), 0.2 mg biotin, 3 mg folic acid, 375 mg ferrous sulfate monohydrate, 77.6 mg Mn, 131 mg Cu oxide, 80.3 mg Zn oxide, 1.5 mg I and 0.3 mg Se. ^2^ Nutrient and digestible energy content was calculated using Plurimix software (Fabermatica, CR, Italy).

**Table 2 animals-12-00203-t002:** Growth performance of piglets fed the control diet (CON) or diet supplemented with *Passiflora incarnata* powder extract (PAS) standardized for vitexin.

	n	CON	PAS	SEM	*p*-Value
Initial BW, kg	60	9.13	9.01	0.21	0.779
Final BW, kg	56	22.48	22.49	0.43	0.989
ADG, g/day	56	0.475	0.485	0.01	0.654
FCR, kg/kg	-	2.48	2.44	0.027	0.418

ADG: average daily gain; BW: body weight; FCR: feed conversion ratio; SEM: pooled standard error of the means.

**Table 3 animals-12-00203-t003:** Descriptive statistics of temperatures (mean ± SE) measured in the eye, ear (front and back position) and dorsal area in two experimental groups (CON and PAS).

	CON	PAS	Average	*p*-Value	
				Diet	Time
T (°C) eye					
d-1	37.72 ± 0.10	37.71 ± 0.08	37.71		
d-10	37.82 ± 0.10	37.87 ± 0.08	37.84		
d-19	38.48 ± 0.08	38.07 ± 0.09	38.27		
d-28	38.04 ± 0.14	38.05 ± 0.07	38.04		
				ns	*p* < 0.01
T (°C) ear back					
d-1	38.23 ± 0.08	38.52 ± 0.10	38.37		
d-10	38.29 ± 0.09	38.61 ± 0.09	38.45		
d-19	39.08 ± 0.10	38.62 ± 0.07	38.85		
d-28	38.74 ± 0.17	38.48 ± 0.11	38.61		
				ns	*p* < 0.01
T (°C) ear front					
d-1	38.19 ± 0.09	38.38 ± 0.15	38.28		
d-10	38.42 ± 0.10	38.71 ± 0.07	38.56		
d-19	39.22 ± 0.12	38.77 ± 0.15	38.99		
d-28	38.88 ± 0.11	38.94 ± 0.11	38.91		
				ns	*p* < 0.01
T (°C) dorsal					
d-1	37.05 ± 0.10	37.09 ± 0.12	37.07		
d-10	37.04 ± 0.13	37.38 ± 0.11	37.21		
d-19	38.20 ± 0.09	37.75 ± 0.09	37.97		
d-28	37.65 ± 0.16	37.71 ± 0.10	37.68		
				ns	*p* < 0.01

ns—not significant (*p* ≥ 0.05); SE—standard error; CON: control animals receiving only the basal diet without natural extract supplementation; PAS: animals receiving the control diet supplemented with *Passiflora incarnata* extract (1 kg/t).

**Table 4 animals-12-00203-t004:** Salivary IgA concentrations expressed as Log in the two experimental groups.

	CON	PAS		*p*-Value	
			Diet	Time	Diet × Time
day 1	1.47	1.80			
day 10	1.52	1.56			
day 19	1.53	1.55			
day 28	1.61	1.46			
			ns	ns	*p* < 0.001

CON: control animals receiving only the basal diet without natural extract supplementation; PAS: animals receiving the control diet supplemented with *Passiflora incarnata* extract (1 kg/t); ns: not significant (*p* ≥ 0.05).

**Table 5 animals-12-00203-t005:** Total occurrence of selected behaviors, expressed as Log, in piglets fed the control diet (CON) or diet supplemented with *Passiflora incarnata* powder extract (PAS).

	CON	PASS	*p*-Value		
			Diet	Time	Diet × Time
Aggression					
days 7–9	1.815	1.582			
days 13–15	1.814	1.466			
days 20–22	1.418	0.677			
days 25–27	1.178	0.695			
			*p* < 0.001	*p* < 0.001	*p* < 0.05
Exploration of enrichment					
days 7–9	0.304	1.895			
days 13–15	0.824	1.489			
days 20–22	0.618	1.417			
days 25–27	0.410	1.092			
			*p* < 0.001	*p* < 0.05	*p* < 0.001
Exploration of pen mates					
days 7–9	1.727	1.923			
days 13–15	1.921	1.890			
days 20–22	1.864	1.873			
days 25–27	1.947	1.926			
			Ns	ns	ns
Exploration of tail					
days 7–9	1.624	1.221			
days 13–15	1.380	0.797			
days 20–22	1.317	0.664			
days 25–27	1.142	0.752			
			*p* < 0.001	*p* < 0.001	ns
Exploration of ear					
days 7–9	1.965	1.823			
days 13–15	1.979	1.716			
days 20–22	1.948	1.794			
days 25–27	1.995	1.884			
			*p* < 0.001	*p* < 0.05	*p* < 0.05

ns—not significant (*p* ≥ 0.05).

## Data Availability

The data presented in the current study are available from the corresponding authors on reasonable request.

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
