# Peer review of "Tranquillizing Effect of Passiflora incarnata Extract: Outcome on Behavioral and Physiological Indicators in Weaning Pigs with Intact Tails"

_animals, 2022, doi:10.3390/ani12020203_

Round 1

Reviewer 1 Report

Brief Summary

In their paper „Tranquillizing effect of Passiflora incarnata extract: outcome on behavioural and physiological indicators in weaning pigs with intact tail”, the authors analysed the effect of Passiflora incarnata extract supplemented to the diet in undocked weaning pigs. They focused on the calming effect of the plant extract combined with health data and compared growth performance, IgA, cortisol, temperature as well as behavioural data with a special focus on tail, ear and skin lesions. They indicate, that oral administration of the plant extract results in unchanged growth performance and hypothesised, that among other factors herbal supplementation might have a little calming effect on the animals reducing abnormal behaviour and stress.

General comments

The general language of the article has to be adapted to either AE or BE.

The article handles health and growth comparisons such as IgA and Cortisol and Temperature as well as behavioural comparisons. The readability could be improved, i.e. by presenting the “health” comparisons together, followed by behavioural comparisons. Recently the structure is growth, IgA and Cortisol, behaviour, NOT, Temperature.

The math in the paper should be carefully revised. In L 290 the authors show a “tendentailly significant effect (p=0.055)” which is wrong. An effect ≥ 0.05 is not significant. In Table 4 the authors state as significant <0.05 and >0.05 as not significant it should be ≥ 0.05. In Table 4 the authors probably use an inconsistent decimal delimiter to represent the log-transformed values. Overall the math part could be improved.

In the Results part the authors set a strong focus on the Novel object test (L310) with 2 subfigures and 4 plots. The main result is that all observations are not significantly different in the two dietary groups. In my opinion, this distracts the reader from relevant information and provide no major contributions for the story.

Figure 4 a and b are practically the same b) is in fact 100-a. It is unclear why both plots have to be shown, especially after dichotomising the ear-lesion score. It is uncertain to me, if the same ear-lesion could be counted several times at different time points with different score which then was combined in the dichotomising step.

In the Discussion the authors argue that the higher cortisol levels at d28 indicate a reduced animal welfare and speculate that the higher content in CON vs PAS could be linked to a higher incidence in ear-lesions. That might be reconsidered due to the fact that only d28 show significant different cortisol levels and the ear lesions at d28 are smaller than d19 in the CON group. The authors might consider an increased stress and cortisol levels in the animals due to their increased body size and the reduced available pen size or other more plausible explanations.

The authors distinguish aggressive from exploratory behaviour while considering chewing and biting as exploratory behaviour. It should be clarified if the classes defined are potentially overlapping or distinct from each other and how they differentiate between exploratory chewing and biting from physical contacts.

In general, the authors should consider to be more conservative in interpreting their results about tail and ear exploratory behaviour due to the fact that either the veterinary could remove injured piglets from the pen (L138,139) and the farmer remove the suspected biters from the pen (L474). Consequently the space available to each piglet in the pen increase. However, the authors did not discuss available space yet.

For a reader it is unclear why the results of the not significant NOT between the groups CON and PAS plays such an important role in the Discussion. It does not contribute to the story and should be clarified.

The conclusion should be clarified. The authors state that the high cortisol levels could be linked to incidence of ear-lesions. This is not backed by the data provided. Only at d28 the levels between CON and PAS are significantly different while ear-biting reaches it maximum at d19 in the CON group. In addition the authors state that ear and tail biting is interconnected. In L 526 they state that they did not observe higher frequency in ear-tail biters as they referenced as expected to Brunberg et al. [54].
They discuss furthermore the different aetiology [55] (L480) and show that ear and tail biting has not necessarily to be interconnected. To me as a reader, this is somehow confusing and should be clarified.

The authors did not compare their study to other husbandry procedures which lead to a reduction in tail biting and consequently tail lesions. Therefore, it is unclear how strong the impact of adding Passiflora incarnata extract on tail biting really is and how strong the effect of other procedures, despite docking, on tail-biting is.

Specific comments

 L119 Table 1 should be separated into an Ingredients and a Chemical composition part

L140 please add information about the removal of biters as described in L474, how often an removal happened and how many pens were affected

L176 please provide information about the experience level of the observer

L183 please clarify the meaning of aggressive interaction especially under consideration of the exploration ear/tail classes containing biting and chewing which might also be an aggressive interaction.

L194-209 please clarify why the observer did not use the video system for the NOT which might avoid influencing the animals

L261 correct to Tuckey post-hoc test

L283 correct to (p ≥ 0.05)

L290 p = 0.055 means not significant

L308 Table 4 probably the decimal delimiter is wrong, if not please elaborate

L316-L321 consider removal of Figure 3, due to n.s. effects it does not contribute to the story

L326 correct the figure references

Figure 4: remove either a or b part; one figure instantly provides the information of the other due to the dichotomised scores

Figure 4, Figure 5: suggestion to correct the colour scheme or the diagrams, the reader might be confused by the colours in Fig 4 CON and PAS groups are mentioned and in Fig 5 Tail and Ear Lesions

Figure 5 could be improved by showing tail an ear lesions for each diet

L346 correct to (p ≥ 0.05)

L545-547 please correct: cortisol difference was only observed at d28 whereas the most ear-biting events could be observed at d19

Author Response

Manuscript animals-1513174

Response to Reviewer 1 Comments

In their paper “Tranquillizing effect of Passiflora incarnata extract: outcome on behavioural and physiological indicators in weaning pigs with intact tail”, the authors analysed the effect of Passiflora incarnata extract supplemented to the diet in undocked weaning pigs. They focused on the calming effect of the plant extract combined with health data and compared growth performance, IgA, cortisol, temperature as well as behavioural data with a special focus on tail, ear and skin lesions. They indicate, that oral administration of the plant extract results in unchanged growth performance and hypothesised, that among other factors herbal supplementation might have a little calming effect on the animals reducing abnormal behaviour and stress.

We are grateful for your valuable comments and suggestions, which were very useful to improve our paper. We have carefully revised the manuscript and highlighted all changes. Please find our response on the questions asked below.

General comments

The general language of the article has to be adapted to either AE or BE.

AU: We adapted the language of the whole manuscript to American English. Thanks for your comment.

The article handles health and growth comparisons such as IgA and Cortisol and Temperature as well as behavioural comparisons. The readability could be improved, i.e. by presenting the “health” comparisons together, followed by behavioural comparisons. Recently the structure is growth, IgA and Cortisol, behaviour, NOT, Temperature.

AU: The manuscript was revised according to the comment. Numbering, order and title of paragraphs and sub-paragraphs were charge accordingly in Mat&Met section as well as in Results section. Parameters were merged as “health and physiological indicators”, including Temperature, IgA, Cortisol and lesions, and as “behavioral indicators, including behavioral observation and NOT.

The math in the paper should be carefully revised. In L 290 the authors show a “tendentailly significant effect (p=0.055)” which is wrong. An effect ≥ 0.05 is not significant. In Table 4 the authors state as significant <0.05 and >0.05 as not significant it should be ≥ 0.05. In Table 4 the authors probably use an inconsistent decimal delimiter to represent the log-transformed values. Overall the math part could be improved.

AU: We agree with Rev.1 and “tendentially significant” was replaced with “a trend (p = 0.055) was reported for time” (Lines 299-300) as detailed in the statistical section for value ranging between 0.05 and 0.10 (Lines 259-260). Decimal delimiter and > 0.05 “symbol” in Table 4 (now renamed Table 5) were amended.

In the Results part the authors set a strong focus on the Novel object test (L310) with 2 subfigures and 4 plots. The main result is that all observations are not significantly different in the two dietary groups. In my opinion, this distracts the reader from relevant information and provide no major contributions for the story.

AU: We shortened the text according to your comment (Figure 3 was delete).

Figure 4 a and b are practically the same b) is in fact 100-a. It is unclear why both plots have to be shown, especially after dichotomising the ear-lesion score. It is uncertain to me, if the same ear-lesion could be counted several times at different time points with different score which then was combined in the dichotomising step.

AU: Subfigure b in Figure 4 (now renamed Figure 3) was deleted according to the comment: the percentage of animals not affected by skin lesions at ear area is now presented.

In the Discussion the authors argue that the higher cortisol levels at d28 indicate a reduced animal welfare and speculate that the higher content in CON vs PAS could be linked to a higher incidence in ear-lesions. That might be reconsidered due to the fact that only d28 show significant different cortisol levels and the ear lesions at d28 are smaller than d19 in the CON group. The authors might consider an increased stress and cortisol levels in the animals due to their increased body size and the reduced available pen size or other more plausible explanations.

AU: The effect of reduced space allowance as time increases is equivalent in both groups and adequate compare to law requirements. We hypothesized that the higher cortisol levels period might be linked to an underlying stressful situation that lasted in CON group. The lower incidence of ear lesions (anyway higher compared to PAS) might be the effect of farmer interventions. We add a comment on that (Lines 427-429).

The authors distinguish aggressive from exploratory behaviour while considering chewing and biting as exploratory behaviour. It should be clarified if the classes defined are potentially overlapping or distinct from each other and how they differentiate between exploratory chewing and biting from physical contacts.

AU: We consider two different behavioral categories: agonistic behaviour and exploratory behaviour, identifying the different target of the exploration (pen structures or other pigs). Chewing and biting were considered as specific behavioral elements of exploratory behaviour, while elements for agonistic behaviour were most connected to conflict (fight or displacement with or without physical contact). One of the most recognized hypotheses for the onset of tail biting considers the abnormal behavior as a model of exploration behavior, which becomes dangerous some situations, e.g., lack of environmental stimuli, suckling behaviour connected to early weaning. The paragraph was clarified (Lines 217-223).

In general, the authors should consider to be more conservative in interpreting their results about tail and ear exploratory behaviour due to the fact that either the veterinary could remove injured piglets from the pen (L138,139) and the farmer remove the suspected biters from the pen (L474). Consequently the space available to each piglet in the pen increase. However, the authors did not discuss available space yet.

AU: We discussed the potential influence of farmer’s intervention (i.e. removal of animals and drugs administration) on behavior and lesions (e.g., Lines 506-512). The effect of increased space was equal in both PAS and CON as 2 animals for each group were removed. We clarified that in the text (Lines 267-268).

For a reader it is unclear why the results of the not significant NOT between the groups CON and PAS plays such an important role in the Discussion. It does not contribute to the story and should be clarified.

AU: We shortened the text according to your comment (Lines 513-516).

The conclusion should be clarified. The authors state that the high cortisol levels could be linked to incidence of ear-lesions. This is not backed by the data provided. Only at d28 the levels between CON and PAS are significantly different while ear-biting reaches it maximum at d19 in the CON group. In addition the authors state that ear and tail biting is interconnected. In L 526 they state that they did not observe higher frequency in ear-tail biters as they referenced as expected to Brunberg et al. [54].
They discuss furthermore the different aetiology [55] (L480) and show that ear and tail biting has not necessarily to be interconnected. To me as a reader, this is somehow confusing and should be clarified.

AU: We modified the conclusion giving emphasis to the differences at the different time points, including a possible explanation for higher cortisol content (Lines 526-530). Furthermore, we thank reviewer for the valuable comment. As we do not assess ear biting as abnormal behavior, we modify the text accordingly and we clarify the statement deleting the confusing sentences throughout the text.

The authors did not compare their study to other husbandry procedures which lead to a reduction in tail biting and consequently tail lesions. Therefore, it is unclear how strong the impact of adding Passiflora incarnata extract on tail biting really is and how strong the effect of other procedures, despite docking, on tail-biting is.

AU: We did not address the effect of other husbandry procedures, such as mixing unfamiliar animals, microclimate, health status, as all the recommendations to control tail biting were met. Moreover the housing and management conditions were the same for both groups. We clarified that point in the Mat&Met section (Lines 105-106).

Specific comments

L119 Table 1 should be separated into an Ingredients and a Chemical composition part

AU: We modified the Table 1 according to the comment.

L140 please add information about the removal of biters as described in L474, how often an removal happened and how many pens were affected

AU: We clarified the information about the removed animals (Lines 267-268).

L176 please provide information about the experience level of the observer.

AU: The observer was trained by checking repeatability of video scoring for behavioral observation and of direct scoring for the NOT.

L183 please clarify the meaning of aggressive interaction especially under consideration of the exploration ear/tail classes containing biting and chewing which might also be an aggressive interaction.

AU: See the previous answer. The paragraph was clarified (Lines 217-223).

L194-209 please clarify why the observer did not use the video system for the NOT which might avoid influencing the animals.

AU: We acknowledge that video recording do not affect the behavior of animals. Nevertheless, direct observations are more precise in identifying details and allow the observer to “adjust” the frame. This approach is quite useful for NOT test in on-farm setting (in which cameras can have framing constraints) and animals may be hidden behind objects. The operator was trained also to minimized the observer effect.

L261 correct to Tuckey post-hoc test

AU: Thanks for the attention; done (Line 262).

L283 correct to (p ≥ 0.05)

AU: Done as suggested (Line 293).

L290 p = 0.055 means not significant

AU: According to the sentence in Material and methods, paragraph 2.4 (“Alpha level for determination of significance was 0.05 and from 0.05 to 0.10 for trends”) we modified the sentence as follows: “A significant difference (p = 0.008) between the two experimental groups (5.14 ± 0.65 ng/ml CON vs 2.38 ± 0.67 ng/ml PAS) was observed at d-28 and a trend (p = 0.055) was reported for time.” (Lines 298-300).

L308 Table 4 probably the decimal delimiter is wrong, if not please elaborate

AU: We corrected by replacing the comma with the dot as decimal delimiter (Table 4 has now been renamed Table 5).

L316-L321 consider removal of Figure 3, due to n.s. effects it does not contribute to the story

AU: Figure 3 was deleted.

L326 correct the figure references

AU: Done as suggested; thank you for the comment.

Figure 4: remove either a or b part; one figure instantly provides the information of the other due to the dichotomised scores

AU: Done as suggested; we removed part b. (The Figure 4 has now been renamed Figure 3).

Figure 4, Figure 5: suggestion to correct the colour scheme or the diagrams, the reader might be confused by the colours in Fig 4 CON and PAS groups are mentioned and in Fig 5 Tail and Ear Lesions

AU: Done as suggested; we changed the colors of Figure 4 (ex Figure 5).

Figure 5 could be improved by showing tail an ear lesions for each diet

AU: We decided to present the data of tail biting showing the effect of time to underline its significance. We do not present and discuss results of tail and ear lesions by treatment as no significant effect was found.

L346 correct to (p ≥ 0.05)

AU: Done as suggested (Line 338).

L545-547 please correct: cortisol difference was only observed at d28 whereas the most ear-biting events could be observed at d19

AU: We modified the sentence as follows: “The higher cortisol content was observed at the end of trial in control group, whereas the higher proportion of pigs affected by ear lesions was observed at the previous time point. This result might be linked to an underlying stressful situation that lasted during the second half of the experimental period, even showing differences in aggressive behaviors between groups” according to rev. 1 previous comment. (Lines 526-530).

Reviewer 2 Report

The authors' effort to find a solution to tail biting in piglets and to find a substitute method for tail removal is appreciated and they have done a comprehensive study by investigating several related parameters.

I'd like to add a few comments below to upgrade the manuscript.

Introduction- Need to include literature on dietary supplementation of passiflora in animals explaining intake, toxicity or any other adverse effect, palatability etc. You should provide information for the reader in the introduction that Passiflora is a suitable dietary supplementation with no harm to piglets.

Methods- 

Explain more about how did you select littermates, how many sows were taken to collect litter. how many littermates that you have collected from each sow.

Explain more how did you allocate littermates into two treatments groups. did you balance litter mates in between the two treatment groups? and how did you place them in replicate groups of 10. Because this will affect on their behaviours. The mixing protocol of this study is very important for the final conclusion.

diet supplemented with 1 kg/t of Passiflora incarnata- Explain how did you recommend this amount in the diet. Do you have any literature related to the inclusion level?

It seems that the piglets were provided with environmental enrichments when noticed tail biting. I am confused about how did you measure the direct effect of the dietary supplement thereafter? Did you provide additional enrichments to other pens as well? Need further clarification of the procedure here.

Results

Table 4- Better to indicate days for the relevant period: Period 1 ( day 7- day 9), this will provide a better sense to the reader.

Author Response

Manuscript animals-1513174

Response to Reviewer 2 comments

The authors' effort to find a solution to tail biting in piglets and to find a substitute method for tail removal is appreciated and they have done a comprehensive study by investigating several related parameters.

I'd like to add a few comments below to upgrade the manuscript.

Thanks for your effort and time spent on our manuscript; we really appreciate your helpful comments. We have made point to point revision according to your constructive suggestions.

Introduction

Need to include literature on dietary supplementation of passiflora in animals explaining intake, toxicity or any other adverse effect, palatability etc. You should provide information for the reader in the introduction that Passiflora is a suitable dietary supplementation with no harm to piglets.

AU: More details on the suitability and use of Passiflora in pig farming and the information about its inclusion in the register of feed additives were included.

Methods

Explain more about how did you select littermates, how many sows were taken to collect litter. how many littermates that you have collected from each sow.

AU: We include information about the number of litters of origin. Almost all healthy piglets were included (Line 104).

Explain more how did you allocate littermates into two treatments groups. did you balance litter mates in between the two treatment groups? and how did you place them in replicate groups of 10. Because this will affect on their behaviours. The mixing protocol of this study is very important for the final conclusion.

AU: The piglets, balanced for body weight and sex, were divided in 2 experimental dietary groups (5 pens per diet with 12 piglets per pen). The text was amended (Lines 105-106). Concerning the possible effect on behavior, sows were removed and farrowing crates were opened at weaning (average of 26 days). Piglets remained in the farrowing room for another 14 days with the possibility of social interactions among litters. In this way the effect of re-mixing on behavior was equally distributed after selection.

Diet supplemented with 1 kg/t of Passiflora incarnata - Explain how did you recommend this amount in the diet. Do you have any literature related to the inclusion level?

AU: We used the dosage based on a previous work conducted on the piglets (Pastorelli et al 2020). Considering the inclusion of additives, other studies considered the same amount (Fabà et al 2020; Kim et al., 2016; Pastorelli et al 2021).

It seems that the piglets were provided with environmental enrichments when noticed tail biting. I am confused about how did you measure the direct effect of the dietary supplement thereafter? Did you provide additional enrichments to other pens as well? Need further clarification of the procedure here.

AU: All pens were provided with soft wood log and chains from the beginning of the trial (Lines 110-111). As far as the farmer noticed episodes of severe tail biting, he provided additional enrichment materials (shredded paper) to 8 pens (5 pens belonging to CON and 3 pens to PAS), frequently added. We acknowledged that this measure may influence the effect of dietary supplement but it could not be avoided due to ethical reason. Nevertheless, no effect of diet was found.

Results

Table 4- Better to indicate days for the relevant period: Period 1 (day 7- day 9), this will provide a better sense to the reader.

AU: Thanks for the suggestion; we corrected according to the comment.

Reviewer 3 Report

Nice experiment, very well panned, written and conducted.

The fundamental question of the research was to establish if given the therapeutic properties of Passiflora, when included as part of the diet in pigs, it could have a calming effect on them, reducing stress and aggression in animals with intact tails. Which seems to me an interesting and relevant question in the area, given its easy possible application and how cheap it would be.

The subject is original, given that few extracts have been evaluated for this purpose and due to the particular success that this plant has had in other monogastics such as humans.

The paper is well written and easy to read.

The conclusions are consistent with the evidence and arguments presented, and addressed the main question established.

Author Response

Manuscript animals-1513174

Response to Reviewer 3 comments

Nice experiment, very well panned, written and conducted.

The fundamental question of the research was to establish if given the therapeutic properties of Passiflora, when included as part of the diet in pigs, it could have a calming effect on them, reducing stress and aggression in animals with intact tails. Which seems to me an interesting and relevant question in the area, given its easy possible application and how cheap it would be.

The subject is original, given that few extracts have been evaluated for this purpose and due to the particular success that this plant has had in other monogastics such as humans.

The paper is well written and easy to read.

The conclusions are consistent with the evidence and arguments presented, and addressed the main question established.

We thank the positive appraisal from our reviewer on our work and the manuscript itself.
